# Cognitive Insight in First-Episode Psychosis: Changes during Metacognitive Training

**DOI:** 10.3390/jpm10040253

**Published:** 2020-11-27

**Authors:** Irene Birulés, Raquel López-Carrilero, Daniel Cuadras, Esther Pousa, Maria Luisa Barrigón, Ana Barajas, Ester Lorente-Rovira, Fermín González-Higueras, Eva Grasa, Isabel Ruiz-Delgado, Jordi Cid, Ana de Apraiz, Roger Montserrat, Trinidad Pélaez, Steffen Moritz, Susana Ochoa

**Affiliations:** 1Parc Sanitari Sant Joan de Déu, Sant Boi de Llobregat, 08830 Barcelona, Spain; i.birules@pssjd.org (I.B.); raquellopez@pssjd.org (R.L.-C.); d.cuadras@pssjd.org (D.C.); aapraiz@pssjd.org (A.d.A.); rm.jovellar@pssjd.org (R.M.); tpelaez@pssjd.org (T.P.); 2Department of Cognition, Development and Educational Psychology, Universitat de Barcelona, 08035 Barcelona, Spain; 3Investigación Biomédica en Red de Salud Mental (CIBERSAM) Instituto de Salud Carlos III C/Monforte de Lemos 3-5, Pabellón 11, Planta 0, 28029 Madrid, Spain; esterlorente@hotmail.com (E.L.-R.); egrasa@santpau.cat (E.G.); 4Fundació Sant Joan de Déu, Esplugues de Llobregat, Santa Rosa, 39-57, 3a planta 08950 Esplugues de Llobregat, Barcelona, Spain; 5Institut de Recerca en Salut Mental Sant Joan de Déu, Parc Sanitari Sant Joan de Déu, Sant Boi de Llobregat, 08830 Barcelona, Spain; 6Department of Psychiatry, Hospital de la Santa Creu i Sant Pau, 08041 Barcelona, Spain; epousa@santpau.cat; 7Consorci Corporació Sanitària Parc Taulí de Sabadell, Parc Taulí, 1, 08208 Sabadell, Barcelona, Spain; 8Psychiatry Service, Area de Gestión Sanitaria Sur Granada, Motril, 18600 Granada, Spain; marisabe@gmail.com; 9Department of Psychiatry, IIS-Fundación Jiménez Díaz Hospital, 28040 Madrid, Spain; 10Centre d’Higiene Mental Les Corts, 08029 Barcelona, Spain; ana.barajas@chmcorts.com or; 11Departament de Psicologia Clínica i de la Salut, Facultat de Psicologia, Universitat Autònoma de Barcelona, Bellaterra, 08193 Barcelona, Spain; 12Psychiatry Service, Hospital Clínico Universitario de Valencia, 46010 Valencia, Spain; 13UGC Salud Mental de Jaén, Servicio Andaluz de Salud, 23007 Jaen, Spain; pablofermingh78@gmail.com; 14Institut d’Investigació Biomèdica-Sant Pau (IIB-Sant Pau), Universitat Autònoma de Barcelona, 08193 Barcelona, Spain; 15Unidad de Salud Mental Comunitaria Málaga Norte, UGC Salud Mental Carlos Haya, Servicio Andaluz de Salud Psychiatry Service, Antequera, 29200 Málaga, Spain; isabelruizdelgado@hotmail.com; 16Mental Health & Addiction Research Group, IdiBGi, Institut d’Assistència Sanitària, 17190 Girona, Spain; jordi.cid@ias.cat; 17Department of Psychiatry and Psychotherapy, University Medical Center Hamburg, 20251 Hamburg, Germany; moritz@uke.uni-hamburg.de; 18Sant Boi de Llobregat, 08830 Barcelona, Spain

**Keywords:** first-episode psychosis, metacognitive training, cognitive insight, sessions, experiment

## Abstract

Background: Metacognitive training (MCT) has demonstrated its efficacy in psychosis. However, the effect of each MCT session has not been studied. The aim of the study was to assess changes in cognitive insight after MCT: (a) between baseline, post-treatment, and follow-up; (b) after each session of the MCT controlled for intellectual quotient (IQ) and educational level. Method: A total of 65 patients with first-episode psychosis were included in the MCT group from nine centers of Spain. Patients were assessed at baseline, post-treatment, and 6 months follow-up, as well as after each session of MCT with the Beck Cognitive Insight Scale (BCIS). The BCIS contains two subscales: self-reflectiveness and self-certainty, and the Composite Index. Statistical analysis was performed using linear mixed models with repeated measures at different time points. Results: Self-certainty decreased significantly (*p* = 0.03) over time and the effect of IQ was negative and significant (*p* = 0.02). From session 4 to session 8, all sessions improved cognitive insight by significantly reducing self-certainty and the Composite Index. Conclusions: MCT intervention appears to have beneficial effects on cognitive insight by reducing self-certainty, especially after four sessions. Moreover, a minimum IQ is required to ensure benefits from MCT group intervention.

## 1. Introduction

Schizophrenia and first-episode psychosis represent one of the most invalidating disorders. It concurs with high psychosocial disability and is associated with stigma and discrimination. Although it has a strong genetic basis, psychosocial protective aspects must always be considered to ensure their acceptance and integration in the community [1,2]. In this line, psychological aspects in relation to symptoms should be explored. The relationship between cognitive biases and psychotic symptoms has been well studied. People with psychosis are more likely to present some characteristic cognitive biases such as personal attributional style [3,4], self-serving bias [5], bias against disconfirmatory evidence [6], and jumping to conclusions [7,8]. Apart from cognitive distortions, it has also been found that people suffering from psychosis have more difficulties in social cognition, in particular in theory of mind (ToM) [9] and emotional recognition [10].

Metacognitive training (MCT) was developed in order to deal with the problems related to cognitive biases and social cognition in psychosis [11]. Metacognition is broadly defined as cognition about one’s own cognitions [12]. MCT consists of a manualized group training program of eight sessions addressed to reduce cognitive biases that are putatively involved in the formation and maintenance of psychotic symptoms such as jumping to conclusions and overconfidence in errors; hence, MCT is designed to improve social cognition. Previous research indicates that MCT is an effective psychological intervention for people with schizophrenia [13]. Specifically, in recent-onset of psychosis, MCT is an effective treatment for improving psychotic symptoms, cognitive insight, and attributional style, as well as for reducing irrational beliefs [14]. Moreover, people who attend MCT deemed it positive in terms of entertainment and usefulness in everyday life, and most of them will recommend it [11,15,16].

The present study builds on a previously published controlled trial assessing the efficacy of MCT in a recent-onset of psychosis sample. A remarkable observation from this study was that the MCT group had a significantly higher improvement in cognitive insight when compared with the psycho-educational group [14]. Cognitive insight refers to the cognitive processes involved in the metacognitive ability to examine and question one’s beliefs and appraisals and to re-evaluate anomalous experiences and misinterpretations. It should be differenced from clinical insight (or unawareness of insight) and anosognosia, considering these concepts as a lack of awareness of symptoms, their need for treatment, and its consequences in daily life or problems in basic cognitive processes [17]. The absence of insight occurs by a failure of objectivity, a loss of ability to put this into perspective, a resistance to correct information from the other opinions, and an excess of confidence in the conclusions. Cognitive insight, usually assessed with the Beck Cognitive Insight Scale (BCIS), includes two subscales: self-certainty and self-reflectiveness [18,19]. Higher self-certainty scores reflect greater confidence about being right and more resistance to correction, while higher self-reflectiveness scores indicate the willingness to question one’s thoughts and greater capacity to analyze them with perspective. Increasing cognitive insight is associated with higher levels of metacognition and fewer symptoms in people with first-episode psychosis [20]. Moreover, a positive relationship between cognitive insight and premorbid intelligence quotient (IQ) and educational level has been described in several studies [21]. MCT enhances the ability to distance oneself from one’s own thoughts and misinterpretations and to reappraise them. Reducing self-certainty is one of the core aims of MCT. By reducing this attitude, the overconfidence bias in one’s own thoughts is reduced and, presumably, the risk of emergence of new delusions is also decreased [22]. Furthermore, a recent study shows cognitive insight training can improve meaning-making in patients and help them come to terms with their diagnosis [23].

As MCT is known to improve cognitive insight, it would be of interest to analyze the results of each subcomponent separately and the effect of each session on self-reflectiveness and self-certainty. To the best of our knowledge, there is no published study analyzing the effects of each MCT session in cognitive insight. Therefore, the aim of this study was to assess the changes in cognitive insight before and after treatment, and then after the follow-up measures and after each session of the MCT intervention, controlling for IQ and educational level.

Our hypothesis is that the aforementioned changes will progressively increase throughout treatment and will be maintained at follow-up. We expect that significant changes will be detectable after two or three sessions because of the complexity of the nuclear construct in the formation of delusions.

## 2. Materials and Methods

### 2.1. Participants

Patients with recent-onset of psychosis were recruited by staff members of the nine participating Spanish mental health centers. A total of 126 patients were recruited and 122 cases were finally analyzed, as 4 patients did not continue the study after enrollment.

Patients were randomized during inclusion into the experimental or psycho-educational group by blocks of four from the list of random numbers in each center’s MCT group. In the present study, the data from the 65 patients belonging to the experimental group (44 men and 21 women) were analyzed. Of this group, 48 completed the treatment and were evaluated in the post-treatment, and 17 were lost during the intervention, declining to participate. Forty-one patients completed the study, as 7 patients discontinued the treatment.

The inclusion criteria were as follows: (1) presence of one of the following diagnoses (according to DSM-IV-TR): schizophrenia, psychotic disorder not otherwise specified, delusional disorder, schizoaffective disorder, brief psychotic disorder, schizophreniform disorder; (2) less than five years from the onset of symptoms; (3) score during the previous year ≥3 in item delusions, grandiosity, or suspicions of Positive and Negative Syndrome Scale (PANSS) positive subscale; and (4) age between 17 and 45. The exclusion criteria were as follows: (1) traumatic brain injury, dementia, or intellectual disabilities (premorbid IQ ≤ 70); (2) substance dependence; and (3) scores on the PANSS ≥ 5 in hostility and uncooperative and ≥6 in suspiciousness, in order to avoid altering the dynamics of the group.

### 2.2. Instruments

Patients were assessed by a blinded evaluator at baseline, after each session, post-treatment, and at six months follow-up. The evaluators were trained in the scales of the study, scoring > 0.70 in inter-rater reliability. Assessment included clinical, meta-cognitive, social, and neuropsychological functioning [14]. For the present study, only the data from the Beck Cognitive Insight Scale were analyzed. By administering the BCIS scale [18,19] after each session, at baseline, at post-treatment, and at 6 months follow-up, the correction of distorted beliefs and misinterpretations through scores in the self-reflectiveness and self-certainty subscales could be evaluated.

Beck Cognitive Insight Scale (BCIS) is a self-administered scale assessing cognitive insight yielding a nine-item self-reflectiveness subscale and a six-item self-certainty subscale, as well as a Composite Index score. The Composite Index is calculated by subtracting the score for the self-certainty scale from that of the self-reflectiveness scale. Higher scores in self-reflectiveness and the Composite Index indicate higher cognitive insight, while low scores in self-certainty indicate better cognitive insight. Sensibility and specificity values of the scale are described by Martin et al. [24]. Respondents are asked to rate how much they agree with each statement using a four-point scale that ranges from 0 (do not agree at all) to 3 (agree completely). Self-report instrument was constructed to contain two sets of items. The first set included items relevant to objectivity, reflectiveness, and openness to feedback. The questions were written to capture patients’ recognition that they could be wrong even when they felt strongly that they were right, that other people could be more objective than they were, and that they were willing to consider other people pointing out that their beliefs were wrong. An item was included to evaluate the patients’ acceptance of the notion of alternative explanations. There were also items about patients being receptive to feedback, being able to make more adaptive attributions, and being able to admit to inadequate cognitive strategies. Perspective was based on the recognition that patients had misconstrued peoples’ attitudes towards themselves, that they had jumped to conclusions too fast, that certain experiences that had seemed real were due to their imagination, that some of the ideas they believed to be true were false, and that some of their unusual experiences were due to their being upset or stressed. The second set of items in the BCIS was written to address decision-making regarding mental products: jumping to conclusions, certainty about being right, and resistance to correction. These six items addressed patient’s certainty about their beliefs and conclusions, such as doing something if it feels right, dogmatic rightness, and resistance to feedback from others [18]. The internal consistency coefficients (Cronbach’s alpha) of the Spanish adaptation of BCIS for schizophrenia were 0.59 for self-reflectiveness and 0.62 for self-certainty [19].

Premorbid intelligence quotient (IQ) was estimated with the Vocabulary Subtest of the Spanish adaptation of the Wechsler Adult Intelligence Scale (WAIS-III) [25,26].

Patients were assessed with a sociodemographic and clinical questionnaire created ad hoc at baseline. We collected data on gender, academic background, cohabitants, and age of onset of the disease.

### 2.3. Procedure

As mentioned above, this study builds on a previous multicenter randomized clinical trial in which one group received MCT, while the control group was a psycho-educational group matched in frequency and duration. Both interventions were applied in a group setting.

The current study focused on the results of insight scores only in the metacognitive group. The research process flow-chart is shown in Figure 1.

The project was evaluated by the research and ethics committees of each participating center in the study. The first evaluation was performed by Sant Joan de Déu Ethics Committee (PIC-73-11). The participants were informed about the aims of the study and signed informed consent for participation in the study. The main study was registered in the Clinical Trials registry (Identifier: NCT02340559).

The intervention consisted of eight weekly group sessions of MCT. All therapists were trained by Steffen Moritz, author of MCT, and Lisa Schilling. The MCT program included eight modules (one for each session): module 1: Avoid the only causes and uncontrolled; modules 2 and 7: Jumping to conclusions; module 3: Cognitive flexibility; modules 4 and 6: Theory of mind; module 5: Overconfidence in memory errors; and module 8: Depression and low self-esteem.

The software used was R 3.0 (R Foundation for Statistical Computing, Vienna, Austria). Statistical analysis was performed using linear mixed models with repeated measures at different time points, with the baseline evaluation as the reference. The dependent variables are the two BCIS subscales and the Composite Index (a model is estimated for each of them individually), and the IQ and educational level were included as covariates.

Two analyses were performed. First, three time points were used: the baseline assessment, the post-treatment assessment, and the six months follow-up. Secondly, the results after each session and at the end of the treatment were used in order to determine whether there were MCT sessions that were more effective than others in improving cognitive insight.

Empirical size effects (Cohen’s d) were also calculated, comparing each evaluation with the baseline. These analyses used all the available data without the need to impute missing values.

## 3. Results

Table 1 shows the sociodemographic characteristics of the sample.

Average attendance in the training program was 5.53/8 sessions (SD = 2.46) in session 1: 50 patients, session 2: 42, session 3: 28, session 4: 37, session 5: 41, session 6: 36, session 6: 36, and session 8: 40. No differences were found in the linear mixed model regarding the number of sessions attended and educational level, so these variables were not controlled for. The results were controlled for intellectual quotient as a significant influence on insight was found.

Self-certainty decreased significantly (*p* = 0.03) over time and the effect of IQ was negative and significant (*p* = 0.02) (patients with higher IQ obtained lower scores in self-certainty). Changes in self-reflectiveness throughout the treatment and at the six months follow-up were far from significant (*p* = 0.99), but IQ had a positive effect. Those with a higher IQ obtained better results in self-reflectiveness (*p* < 0.01). The Composite Index increased over time, but the coefficient was not significant (*p* = 0.36). The effect of IQ was significantly positive (*p* < 0.01), so that patients with higher IQ obtained better results in the Composite Index.

Figure 2, Figure 3 and Figure 4 show the scores of BCIS subscales in each session.

Table 2 shows the results regarding the effect of changes in self-certainty, self-reflectiveness, and the Composite Index in each session. From session 4 to session 8 (ToM, overconfidence in memory errors, jumping to conclusions, and depression and self-esteem), all sessions improved cognitive insight by significantly reducing self-certainty. In the post-treatment assessment, changes in cognitive insight were maintained. No differences by session were found regarding the self-reflectiveness BCIS subscale. However, a trend towards significance was found regarding the influence of IQ. Regarding the Composite Index, from session 4 to session 7 (theory of mind, overconfidence in memory errors, and jumping to conclusions), significant improvements in cognitive insight were found. In session 8 (depression and self-esteem), a trend towards significance was found. These improvements were maintained post-treatment. IQ was significant in the analysis.

## 4. Discussion

As expected, the results revealed that cognitive insight improved with the MCT, in particular regarding the self-certainty subscale. These results were maintained at the six months follow-up. Regarding the effect of each session in changing cognitive insight, the results indicated that, after the fourth session, there was an improvement in cognitive insight, particularly in the self-certainty subscale and in the Composite Index. Moreover, it was found that IQ could have an effect on insight changes, revealing that people with a higher IQ improved more in insight.

Patients included in the MCT improved their cognitive insight over the course of treatment, in particular in the self-certainty domain, coinciding with other studies [27]. It is likely that different mechanisms are involved. Analysing the effect of each session, the results show a significant improvement in cognitive insight after session 4 (ToM). These results suggest that there is a cumulative effect over the sessions. Despite this suggestion, not all the sessions seemed to be equally efficient; some of them had a bigger effect size: ToM (session 4), memory (session 5), and empathy (session 6) are the sessions with the greatest contribution to reducing self-certainty. Considering that self-certainty is intended to reduce dogmatic rightness and resistance to feedback from others, these sessions were the ones that best achieved this aim. Session 4 (ToM) and session 6 (empathy) were the two sessions focused on working on relationships with the others and the idea that others can have a different point of view from us, facilitating patients to have more doubt about the certainty of their own thoughts. Session 5 (memory) tackled the memory errors that everyone can experience in everyday life such as rebuilding memories with mistakes. The results of the Composite Index coincide with the self-certainty results, except for the eighth session, where a trend towards significance was found. These results have some clinical implications. Benoit et al. found that better scores in self-certainty were related to improvement in speed processing and visual memory in people attending a remediation cognitive program [28]. Other studies have related self-certainty insight with premorbid IQ, premorbid academic adjustment, and clinical insight in patients with first-episode psychosis [21], as well as with jumping to conclusion and weaknesses in cognitive flexibility, assessed with the Wisconsin Card Sorting Test (WSCT), in At Risk of Mental State subjects [29,30].

Overall, no significant improvements were found regarding self-reflectiveness during the course of the MCT group. Our results are contradictory to those found by Lam et al. performed in Chinese people with schizophrenia, which found significant improvements in cognitive insight and increasing self-reflectiveness, but not self-certainty in those patients included in the MCT condition [15]. There are some differences with our study; Lam’s study was done with people with schizophrenia, included inpatients and outpatients, compared with treatment as usual (TAU), was performed twice a week, and was delivered by occupational therapists. On the other hand, patients from our sample scored higher in self-reflectiveness at baseline compared with other studies of validation of the instrument and of intervention [15,31], which suggests a ceiling effect for the self-reflectiveness subscale. Moreover, it should be taken into account that other authors have not found differences in the self-reflectiveness subscale between healthy and psychotic populations [22,24]. Additionally, the Composite Index improved in the same sessions as self-certainty did. Moreover, it should be considered that the biggest effect size was found in session 3 (cognitive flexibility), although it was not statistically significant, probably because of the small number of participants in this session.

Considering the different studies regarding this issue, the results obtained are different in both dimensions of the cognitive insight. In this line, this finding suggests separately studying the sub-components of cognitive insight because higher cognitive insight does not always lead to the same psychological functioning. For instance, higher levels of self-reflectiveness are often associated with depressive mood [32] and with functional capacity [33], while self-certainty has been related to cognitive function [21,23]. Considering the different effect of each dimension, it should be taken independently and assessed at baseline in order to better fit the intervention. Moreover, the Composite Index has been related to social functioning [34]. In this line, it would be useful to implement the MCT intervention in first-episode patients before psychosocial therapies.

In the main part of our study, the results suggest that self-reflectiveness was significantly better in the follow-up regarding the MCT group than in the control group. Self-reflectiveness scores decreased, while in the MCT group, they were stable [14], considering a possible sleeper effect of the MCT intervention as suggested by Moritz, Veckenstedt, et al. in a 3-year follow-up study [35] and Sarin et al. in Cognitive Behavioral Therapy [36]. In the present study, no differences were found in self-reflectiveness over time, but it did not decrease as it did in the control group. The increase in self-reflectiveness and consequently in cognitive insight could act as a protective factor against future psychotic episodes and as a better functioning response, as suggested by Benoit et al. [33].

Intellectual quotient (IQ) is related to cognitive insight changes; patients with a higher IQ are more likely to improve their cognitive insight (concretely in the Composite Index and a tendency in self-reflectiveness). These results are in concordance with those of González-Blanch et al. [21] and with the meta-analysis of Nair et al. [37]. So, taking into consideration this result, it is recommended to guarantee a minimum IQ before considering patients for MCT intervention in order to maximize benefit.

Some limitations should be considered in the present study. The effect size of the results could be conditioned by the size of the sample included in each session. Not all the patients attended every session and some of them had lower attendance rates. Another limitation is that the cumulative effect of treatment is not controlled because all the patients received the sessions in the same order. However, there was no progressive increase in the effect size, suggesting that some sessions are more useful than others. Future studies should consider a randomization of the sessions in order to avoid the learning effect and to control the real effect of each session independently. The main study was performed before the launch of the DSM V. Therefore, patients were diagnosed according to DSM-IV-TR criteria. Of note, the remarkable difference between the two editions concerns subtypes of schizophrenia. Subtypes of schizophrenia was not a variable of our study. Furthermore, although the main project compared the effectiveness of MCT over a psychoeducational group, the BCIS in each session was assessed only in the MCT group. Future researchers should assess BCIS in both groups in order to compare them in terms of cognitive insight.

A first clear implication of our results is the importance of psychological interventions, such as metacognitive training, in addressing cognitive biases in first-episode psychosis. In daily clinical practice, it is sometimes unrealistic to administer complete interventions to all the patients. Therefore, we believe that the following sessions of the MCT should be a priority: attributional style, memory mistakes, and stimulating empathy and theory of mind. These modules are more implicated in reducing self-certainty and in increasing thought flexibility. In turn, improvements in these areas could prevent the formation of delusions and a new relapse. Considering the present health emergency due to the COVID-19 pandemic, adapting its application on virtual settings emerges as a new challenge.

In conclusion, MCT could be considered an appropriate psychological intervention to improve cognitive insight. The study has demonstrated that, after the fourth session, improvements in insight are evident in those patients who received the MCT intervention. Moreover, IQ is an important component to take into account before initiating the MCT intervention.

## Figures and Tables

**Figure 1 jpm-10-00253-f001:**
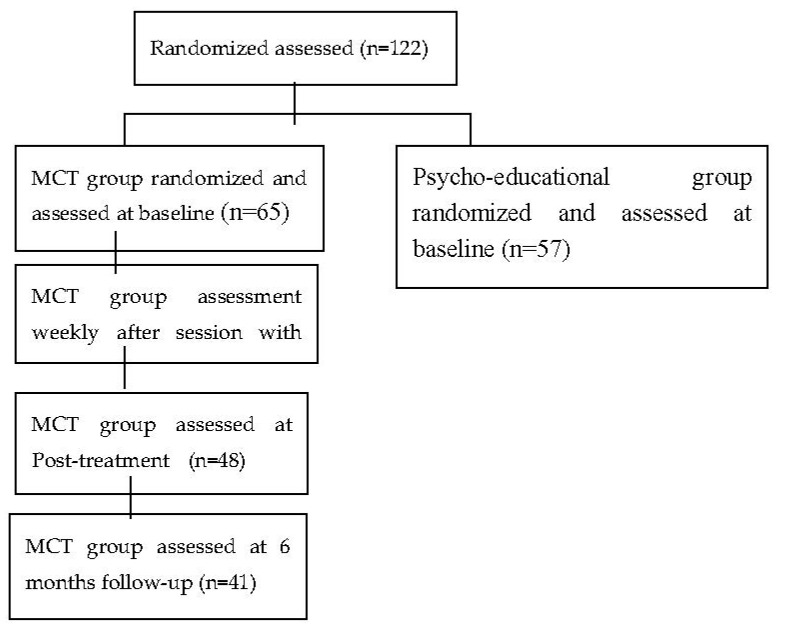
Research process flow-chart. MCT, metacognitive training.

**Figure 2 jpm-10-00253-f002:**
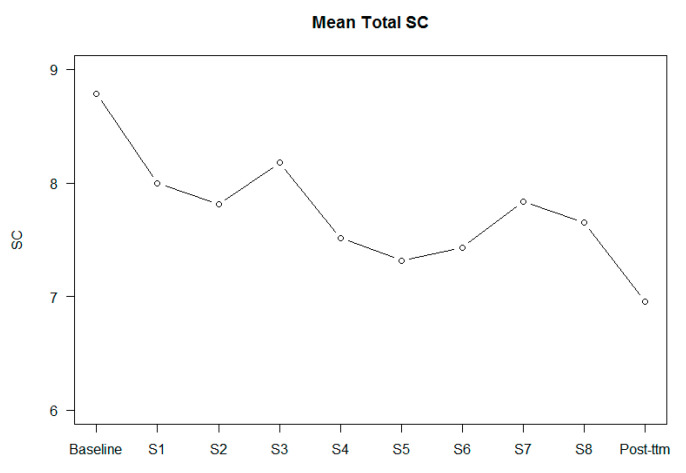
Changes in self-certainty (SC) in every session and in the post-treatment.

**Figure 3 jpm-10-00253-f003:**
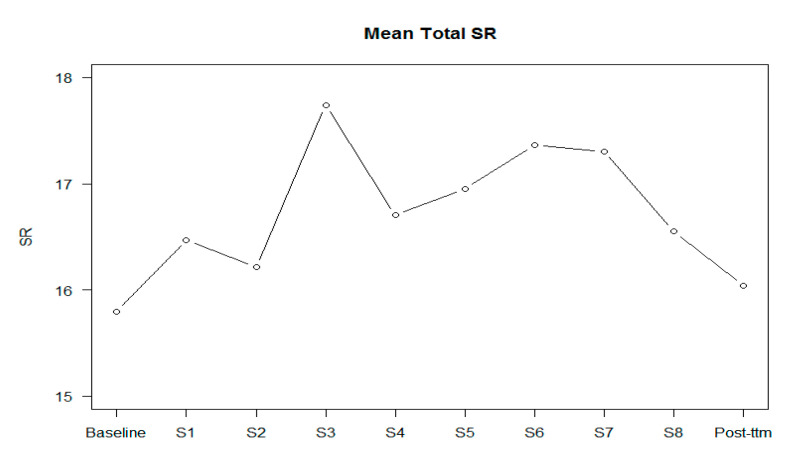
Mean of self-reflectiveness (SR) in each session and in the post-treatment.

**Figure 4 jpm-10-00253-f004:**
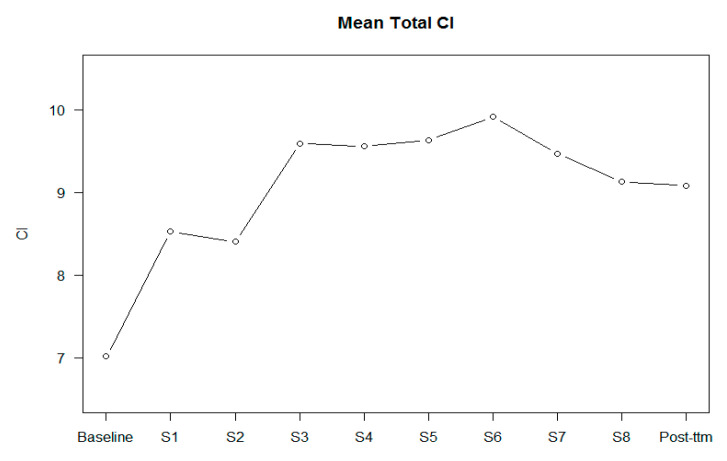
Mean of the Composite Index (CI) in each session and in the post-treatment.

**Table 1 jpm-10-00253-t001:** Socio-demographic characteristics of the sample. MCT, metacognitive training.

		MCT Group	
		*N*	*%*
Gender	Men	44	67.7
	Women	21	32.3
Marital status	Single	53	81.5
	Married	8	12.3
	Divorced	4	6.2
Level of education	Primary	26	40.0
	Secondary	25	38.5
	University	14	21.5
Employment status	Working	14	21.5
	Student	12	18.5
	Incapacity	13	20.0
	Unemployed	19	29.3
	Other	7	10.7
		Mean	SD *
Age		27.05	7.94
Age at onset		25.16	7.79
Years of psychosis duration		2.15	2.01
Number of hospitalizations		1.16	1.54
Antipsychotic dose mg/d **		472.53	703.89

* SD = Standard Deviation. ** Antipsychotic drug doses are expressed as chlorpromazine equivalence.

**Table 2 jpm-10-00253-t002:** Effects of each session and at the post-treatment in self-certainty, self-reflectiveness, and the Composite Index and influence of intelligence quotient (IQ).

	Self-Certainty	Self-Reflectiveness	Composite Index
	Value	Std. Error	*t*-Value	*p*-Value	Effect Size	Value	Std. Error	*t*-Value	*p*-Value	Effect Size	Value	Std. Error	*t*-Value	*p*-Value	Effect Size
Intercept	12.181	2.448	4.976	<0.001		10.280	3.108	3.308	0.001		−1.432	4.276	−0.335	0.738	
S1	−0.606	0.395	−1.532	0.127	0.126	0.216	0.534	0.405	0.685	0.064	0.853	0.655	1.303	0.194	0.147
S2	−0.771	0.421	−1.832	0.068	0.112	0.027	0.564	0.048	0.962	0.046	0.832	0.693	1.201	0.231	0.116
S3	−0.785	0.488	−1.608	0.109	0.111	0.311	0.664	0.468	0.640	0.283	1.260	0.815	1.546	0.123	0.326
S4	−1.404	0.449	−3.124	0.002	0.380	0.020	0.591	0.034	0.973	0.013	1.621	0.747	2.170	0.031	0.265
S5	−1.294	0.419	−3.091	0.002	0.364	0.630	0.561	1.123	0.262	0.036	1.956	0.689	2.839	0.005	0.251
S6	−1.357	0.449	−3.020	0.003	0.309	0.657	0.596	1.101	0.272	0.146	2.156	0.739	2.916	0.004	0.321
S7	−1.181	0.441	−2.678	0.008	0.210	0.567	0.591	0.961	0.338	0.085	1.778	0.725	2.451	0.015	0.202
S8	−0.946	0.425	−2.225	0.027	0.242	0.237	0.580	0.410	0.682	0.016	1.346	0.712	1.891	0.060	0.279
Post-treatment	−1.564	0.399	−3.920	0.000	0.460	−0.031	0.535	−0.059	0.953	0.062	1.570	0.656	2.392	0.017	0.189
IQ	−0.035	0.025	−1.404	0.166		0.062	0.032	1.949	0.056		0.092	0.044	2.107	0.039

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
