# Peer review of "Cognitive Insight in First-Episode Psychosis: Changes during Metacognitive Training"

_jpm, 2020, doi:10.3390/jpm10040253_

Round 1
Reviewer 1 Report
The manuscript investigates a relevant issue that is impact of metacognitive training on cognitive insight in first-episode psychosis.
In an original research, the Authors detail: a) brief description of the reference context (metacognition, Theory of Mind, emotional recognition, cognitive insight); b) research purposes; c) material and methods; d) main results and interpretations.
For what is my concern the topic is relevant and interesting. The scope and the aims of the manuscript are clear and well explained. The content is perfectly in line with the aims of the Journal and enhance the scientific dissemination scope. The chapter is well-written and inferences are clearly outlined. Nevertheless, I suggest some revisions of the manuscript to improve the interpretation of the findings by readers.
Major concerns:
- Please, insert in-depth description of the topic concerning cognitive insight, taking particular care to disambiguate this concept from others often used as synonyms but which are not (especially on the dysfunction side). In particular, a dysfunction of cognitive insight is not anosognosia nor a reduction in self-awareness.
- Please, close with the explanation of the own proper hypotheses. After setting out the aims, it would be good to indicate the questions to be answered and the authors' assumptions.
- The use of the DSM-IV-TR in 2020 poses some limitations to the study. This should be made explicit in the discussion section. I also kindly suggest to indicate the precautions taken in this respect (and any corrective measures) in the section on methods. By way of example, you can refer to Table 3.22DSM-IV to DSM-5 Schizophrenia Comparison (https://www.ncbi.nlm.nih.gov/books/NBK519704/table/ch3.t22/).
- Statistical analyses. Some of the variables presented (e.g. age, level of education, ....) may impact understanding and responsiveness to neuropsychiatric scales. How have these aspects been monitored?
Minor concerns:
- In order to promote the reproducibility of the study, I kindly suggest to indicate the evaluation domains of the "sociodemographic and clinical questionnaire done ad-hoc at baseline".
- I kindly advise the Authors to emphasize their own conclusions by inserting some further reasoning on the future research perspectives and application of MCT in clinical settings. What are the new challenges MCT will face? What are the needs still unheard of?
- Finally, I kindly propose that the authors enrich their manuscript with some infographics:
- a research process flow-chart
- a diagram able to represent the theoretical model suggested by the Authors - in which pathology, cognitive insight, MCT, covariates and explanatory variables detail the possible clinical outcomes.
I thank the authors in advance for their precious collaboration and understanding.
Author Response
Reviewer 1
The manuscript investigates a relevant issue that is impact of metacognitive training on cognitive insight in first-episode psychosis.
In an original research, the Authors detail: a) brief description of the reference context (metacognition, Theory of Mind, emotional recognition, cognitive insight); b) research purposes; c) material and methods; d) main results and interpretations.
For what is my concern the topic is relevant and interesting. The scope and the aims of the manuscript are clear and well explained. The content is perfectly in line with the aims of the Journal and enhance the scientific dissemination scope. The chapter is well-written, and inferences are clearly outlined.
We thank the reviewers for their positive feedback.
Nevertheless, I suggest some revisions of the manuscript to improve the interpretation of the findings by readers.
Major concerns:
- Please, insert in-depth description of the topic concerning cognitive insight, taking particular care to disambiguate this concept from others often used as synonyms but which are not (especially on the dysfunction side). In particular, a dysfunction of cognitive insight is not anosognosia nor a reduction in self-awareness.
We have included this paragraph in the introduction section to clarity the differences between cognitive insight and clinical insight.
“It should be differenced from clinical insight (or unawareness of insight) and anosognosia; considering these concepts as a lack of awareness of symptoms, their need for treatment, and its consequences in daily life or problems in basic cognitive processes.” (Lysaker et al., 2018).
- Please, close with the explanation of the own proper hypotheses. After setting out the aims, it would be good to indicate the questions to be answered and the authors' assumptions.
As suggested by the reviewer, we have included our hypothesis in the manuscript as stated below:
“Our hypothesis is that these
We have included our hypothesis in the manuscript as suggested the reviewer:
“Our hypothesis is that the aforementioned changes will progressively increase throughout treatment and will be maintained at follow-up. We expect that significant changes will be detectable after two or three sessions because of the complexity of the nuclear construct in the formation of delusions”.
- The use of the DSM-IV-TR in 2020 poses some limitations to the study. This should be made explicit in the discussion section. I also kindly suggest to indicate the precautions taken in this respect (and any corrective measures) in the section on methods. By way of example, you can refer to Table 3.22DSM-IV to DSM-5 Schizophrenia Comparison (https://www.ncbi.nlm.nih.gov/books/NBK519704/table/ch3.t22/).
Thank you for pointing out this possible limitation. We have included the following sentence in the limitations section:
“The main study was performed before the launch of the DSM V. Therefore, patients were diagnosed according to DSM-IV-TR criteria. Of note, the remarkable difference between the two editions concerns subtypes of schizophrenia. Subtypes of schizophrenia was not a variable of our study”.
- Statistical analyses. Some of the variables presented (e.g. age, level of education, ....) may impact understanding and responsiveness to neuropsychiatric scales. How have these aspects been monitored?
There was no impact of age and academic background neither in the original version of the BCIS nor in the Spanish validation. Our sample included individuals with first-episode psychosis that were younger and had a high academic background. We expect that these variables did not influence the understanding of the assessment.
We controlled for academic background and IQ. Academic background was not significant in the statistical analysis. A possible reason for it was the high correlation between two variables. We have not included age in the analysis because of the narrow range of scores.
Minor concerns:
- In order to promote the reproducibility of the study, I kindly suggest to indicate the evaluation domains of the "sociodemographic and clinical questionnaire done ad-hoc at baseline".
We have included the following information regarding the sociodemographic and clinical questionnaire.
“Patients were assessed with a sociodemographic and clinical questionnaire created ad-hoc at baseline. We collected data on gender, academic background, cohabitants and age of onset of the disease”.
- I kindly advise the Authors to emphasize their own conclusions by inserting some further reasoning on the future research perspectives and application of MCT in clinical settings. What are the new challenges MCT will face? What are the needs still unheard of?
Thank you for this comment. We have included our own conclusions considering future research and clinical application:
“A first clear implication of our results is the importance of psychological interventions, such as metacognitive training, in addressing cognitive biases in first-episode psychosis. In daily clinical practice, it is sometimes unrealistic to administer complete interventions to all the patients. Therefore, we believe that the following sessions of the MCT should be a priority: attributional style, memory mistakes and, stimulating empathy and theory of mind. These modules are more implicated in reducing self-certainty and in increasing thought flexibility. In turn, improvements in these areas could prevent the formation of delusions and a new relapse. Considering the present health emergency due to the COVID-19 pandemic, adapting its application on virtual settings emerges as a new challenge”.
- Finally, I kindly propose that the authors enrich their manuscript with some infographics:
- a research process flow-chart
A Research process Flow-chart has been added (Figure 1)
- a diagram able to represent the theoretical model suggested by the Authors - in which pathology, cognitive insight, MCT, covariates and explanatory variables detail the possible clinical outcomes.
We have included a diagram representing the theoretical model and research results.(Figure 5)
I thank the authors in advance for their precious collaboration and understanding.

Reviewer 2 Report
This is, in summary, an interesting study aimed to assess changes in cognitive insight after Metacognitive Training (MCT): a) between baseline, post-treatment and follow-up; b) after each session of the MCT controlled for Intellectual Quotient (IQ) and educational level in a total of 65 patients with a first-episode psychosis. The authors found that self-certainty decreased significantly over time and the effect of IQ was negative and significant. In addition, from session 4 to session 8, all sessions improved cognitive insight by significantly reducing Self-certainty and Composite Index.
The authors may find as follows my main comments/suggestions.
First, as throughout the Introduction section, the authors correctly focused on psychotic symptoms, they could further stress the psychosocial disability related to invalidating disorders such as schizophrenia which are wordwide associated with stigma and discrimination. Genetic explanation of schizophrenia may potentially enhance stigma. In particular, considering schizophrenia as a genetic disorder influenced participants perception of other people's beliefs about dangerousness and unpredictability and people's desire for social distance. Importantly, a genetic explanation of schizophrenia was more frequently associated with stigmatizing attitudes. According to a study which has been published in 2013 on J Psychiatr Ment Health Nurs (PMID: 21848591), there were high levels of perceived stigmatization in medical students and medical doctors and at least half of the analyzed subjects perceived stigmatizing social attitudes against psychotic individuals. Thus, given the above information, my additional suggestion is also to rapidly include, throughout the present manuscript, the mentioned paper (PMID: 21848591). Moreover, the genetic liability of disabling conditions like schizophrenia need to be enphasized. In particular, according to a study which has been published in 2014 on World J Biol Psychiatry, gray matter reductions in the anterior cingulate have been reported as markers of genetic liability to psychosis, while reductions in the superior temporal gyrus and cerebellum may be interpreted as markers of a first onset of the illness. Thus, i suggest to briefly cite, within the main text, the specified paper on this specific topic (PMID: 22283467)..
In addition, as the most relevant aims/objectives have been reported extensively, the main study hypotheses of the present study need to be better described in a similarly detailed manner.
Notably, the most relevant psychometric instruments used in the present study might be described more succinctly.
Moreover, the authors should immediately present and discuss, within the first lines of the Discussion section, their most relevant study findings. Conversely, they seem to focus on the main aims/objectives of the present study that should be stressed elsewhere within the main text.
Finally, although the authors reported that after the fourth session, improvements in insight are evident in patients who received the MCT intervention, they failed, in my opinion, to provide some conclusive remarks about their main topic of this investigation. Overall, what is the take-home message of this study? How metacognitive training might be translated in the clinical practice? Here, more details/information are needed for the general readership.
Author Response
Reviewer 2
This is, in summary, an interesting study aimed to assess changes in cognitive insight after Metacognitive Training (MCT): a) between baseline, post-treatment and follow-up; b) after each session of the MCT controlled for Intellectual Quotient (IQ) and educational level in a total of 65 patients with a first-episode psychosis. The authors found that self-certainty decreased significantly over time and the effect of IQ was negative and significant. In addition, from session 4 to session 8, all sessions improved cognitive insight by significantly reducing Self-certainty and Composite Index.
We are grateful for your comments.
The authors may find as follows my main comments/suggestions.
First, as throughout the Introduction section, the authors correctly focused on psychotic symptoms, they could further stress the psychosocial disability related to invalidating disorders such as schizophrenia which are wordwide associated with stigma and discrimination. Genetic explanation of schizophrenia may potentially enhance stigma. In particular, considering schizophrenia as a genetic disorder influenced participants perception of other people's beliefs about dangerousness and unpredictability and people's desire for social distance. Importantly, a genetic explanation of schizophrenia was more frequently associated with stigmatizing attitudes. According to a study which has been published in 2013 on J Psychiatr Ment Health Nurs (PMID: 21848591), there were high levels of perceived stigmatization in medical students and medical doctors and at least half of the analyzed subjects perceived stigmatizing social attitudes against psychotic individuals. Thus, given the above information, my additional suggestion is also to rapidly include, throughout the present manuscript, the mentioned paper (PMID: 21848591). Moreover, the genetic liability of disabling conditions like schizophrenia need to be enphasized. In particular, according to a study which has been published in 2014 on World J Biol Psychiatry, gray matter reductions in the anterior cingulate have been reported as markers of genetic liability to psychosis, while reductions in the superior temporal gyrus and cerebellum may be interpreted as markers of a first onset of the illness. Thus, i suggest to briefly cite, within the main text, the specified paper on this specific topic (PMID: 22283467).
As the reviewer highlights, schizophrenia has a strong genetic base. However, we believe it is important to consider other protective aspects that could help us improve treatments addressed to people with first-episode psychosis and schizophrenia.
We have included the following sentences in the introduction section:
“Schizophrenia and first-episode psychosis are one of the most invalidating disorders. It concurs with high psychosocial disability and is associated with stigma and discrimination. Although it has a strong genetic basis, psychosocial protective aspects must always be considered to ensure their acceptance and integration in the community (Serafini et al., 2011; Fusar-Poli et al., 2014). In this line, psychological aspects in relation to symptoms should be explored”.
In addition, as the most relevant aims/objectives have been reported extensively, the main study hypotheses of the present study need to be better described in a similarly detailed manner.
We agree with your comment. This has also been suggested by reviewer 1, therefore, we have included the following sentence:
“Our hypothesis is that the aforementioned changes will progressively increase throughout treatment and will be maintained at follow-up. We expect that significant changes will be detectable after two or three sessions because of the complexity of the nuclear construct in the formation of delusions”.
Notably, the most relevant psychometric instruments used in the present study might be described more succinctly.
We have included and modified information about the instruments used in the assessment. We have included data on sociodemographic and clinical variables, as it had also been suggested by reviewer 1 too.
Moreover, the authors should immediately present and discuss, within the first lines of the Discussion section, their most relevant study findings. Conversely, they seem to focus on the main aims/objectives of the present study that should be stressed elsewhere within the main text.
Thank you for this comment. We have deleted the first lines of the discussion section and focused in the relevant findings of the study.
Finally, although the authors reported that after the fourth session, improvements in insight are evident in patients who received the MCT intervention, they failed, in my opinion, to provide some conclusive remarks about their main topic of this investigation. Overall, what is the take-home message of this study? How metacognitive training might be translated in the clinical practice? Here, more details/information are needed for the general readership.
We agree with your comment. Considering your and reviewer 1’s suggestions, we have included the following information in the discussion section:
“A first clear implication of our results is the importance of psychological interventions, such as metacognitive training, in addressing cognitive biases in first-episode psychosis. In daily clinical practice, it is sometimes unrealistic to administer complete interventions to all the patients. Therefore, we believe that the following sessions of the MCT should be a priority: attributional style, memory mistakes and, stimulating empathy and theory of mind. These modules are more implicated in reducing self-certainty and in increasing thought flexibility. In turn, improvements in these areas could prevent the formation of delusions and a new relapse. Considering the present health emergency due to the COVID-19 pandemic, adapting its application on virtual settings emerges as a new challenge”.

Round 2
Reviewer 1 Report
I had the opportunity to read carefully the new version of the manuscript and the response comments from the Authors.
I thank the authors for taking into consideration my perplexities and suggestions. Any doubts have been cleared up.
The new version of the manuscript is certainly more exhaustive than the previous one, the presentation of the methods used and the results is clearer, the discussion enriched with important elements. The graphic elements allow a better understanding of the inferences even to an audience of non-experts in the field.
Based on my current skills and knowledge, I suggest the acceptance of the manuscript. I congratulate the authors on the important achievement.
Reviewer 2 Report
In the revised manuscript, the authors addressed most of the major questions raised by Reviewers improving both the main structure and quality of this paper. I have no further additional comments.